# HBeAg-Negative/Anti-HBe-Positive Chronic Hepatitis B: A 40-Year-Old History

**DOI:** 10.3390/v14081691

**Published:** 2022-07-30

**Authors:** Ferruccio Bonino, Piero Colombatto, Maurizia R. Brunetto

**Affiliations:** 1Institute of Biostructure and Bioimaging, National Research Council, Via De Amicis 95, 80145 Naples, Italy; maurizia.brunetto@unipi.it; 2Hepatology Unit and Laboratory of Molecular Genetics and Pathology of Hepatitis Viruses, Reference Center of the Tuscany Region for Chronic Liver Disease and Cancer, University Hospital of Pisa, Via Paradisa 2, 56124 Pisa, Italy; p.colombatto@ao-pisa.toscana.it

**Keywords:** HBV, HBV-DNA, HBeAg, anti-HBe, HBeAg defective HBV mutants, chronic hepatitis B

## Abstract

Hepatitis B “e” antigen (HBeAg) negative chronic hepatitis B (CHB), 40 years since discovery in the Mediterranean area, has become the most prevalent form of HBV-induced liver disease worldwide and a major health care burden caused by HBV infection. A great deal of knowledge accumulated over the last decades provides consistent evidence on the bimodal dynamics of the expression of structural and non-structural forms of the viral core proteins which associate with different virologic and clinic–pathologic outcomes of HBV infection. In absence of serum HBeAg, the presence and persistence of HBV replication causes and maintains virus-related liver injury. Thus, in clinical practice it is mandatory to screen HBV carriers with HBeAg-negative infection for the early diagnosis of HBeAg-negative CHB since antiviral therapy can cure HBV-induced liver disease when started at early stages.

## 1. Discovery

In the mid-1970s, intrahepatic hepatitis B core antigen (HBcAg) was the hallmark of active hepatitis B virus (HBV) replication in carriers of hepatitis B surface antigen (HBsAg) [1]. The HBcAg-positive immunohistochemical staining of the nuclei of HBV infected hepatocytes was indirect evidence of 22 nm viral nucleocapsid particles (core), seen in the electron microscope and associated with HBV-induced inflammatory liver disease [2]. Since its discovery in the serum, hepatitis B “e” antigen (HBeAg) was significantly associated with intrahepatic detection of HBcAg [3,4]. However, a significant proportion of HBeAg-negative/anti-HBe-positive carriers without intrahepatic HBcAg had evidence of unexplained chronic hepatitis, as only a proportion of patients showed hepatitis delta virus (HDV) coinfection [5]. The development of molecular diagnostic assays for detection of serum HBV-DNA, firstly presented at the 1980 AASLD meeting in Chicago, allowed a better understanding of their pathology [6]. Using this technique, our group found that serum HBV-DNA was present in both HBeAg-positive and HBeAg-negative patients with intrahepatic HBcAg [6]. Similar findings were confirmed worldwide after the diffusion of molecular biology techniques for detection of serum HBV-DNA and HBeAg-negative/anti-HBe-positive CHB was shown to prevail in the Mediterranean patients [7,8,9,10]. Serum HBV-DNA levels were lower in HBeAg-negative than HBeAg-positive patients [11,12,13,14]. In addition, a study from our group underlined a peculiar feature of the intrahepatic staining of HBcAg in HBeAg-negative/anti-HBe-positive patients, namely, the concomitant nuclear and cytoplasmic, rather than exclusively nuclear, localization [14]. According to the specific features of HBV infection in anti-HBe-positive patients, their disease was proposed as a distinct clinical entity from HBeAg-positive CHB [15]. Subsequently the study of its virologic characteristics led to the identification of HBeAg defective HBV mutants and their pathogenetic role in HBeAg-negative/anti-HBe-positive CHB [16,17,18,19]. The first study of the dynamic ratios between wild-type HBeAg-positive and HBeAg-negative variant populations within the circulating viral quasispecies of HBeAg-positive carriers revealed that precore G1896A HBeAg-minus HBV variants emerged at the time of hepatitis B exacerbations and were followed by the appearance of circulating antibodies against HBeAg (anti-HBe) [20]. Furthermore, CHB associated with HBeAg-minus HBV infection was characterized by transaminases (ALT) flares intervened with periods of complete ALT normalization: ALT flares were preceded by major increases of viral load that remained very low during the remission phases [20,21]. Altogether, these clinic–pathologic evidences suggested that genetic heterogeneity of the HBV precore region might significantly influence the course and outcome of chronic hepatitis B. The hypothesis was that wild-type HBV favors the induction of chronic infection secreting HBeAg, whereas HBeAg-defective variants, surging after the establishment of chronic infection, selectively prevail, displacing wild-type virus once HBeAg is recognized as immune target, since the lack of HBeAg expression provides a selective advantage to the virus during chronic inflammatory infection (Figure 1). In the last 30 years, a great deal of studies in animals and humans confirmed and further extended this seminal hypothesis. 

## 2. Pathogenesis

HBV is not directly cytopathic, and chronic infection results from virus persistence because of a defective host’s immune response and inability to clear viral infection. A concerted innate and adaptive immune response is thought to be responsible for the control of viral infection and the clearance of intrahepatic necro-inflammation in primary HBV infection [22,23,24]. Humoral antibody response contributes to clearance of circulating virus particles, preventing viral spread, while cellular immune response silences the transcriptional activity of cccDNA by both non-cytolytic pathways and direct infected cell elimination. Anti-HBV T-cell response is vigorous, polyclonal, and multi-specific in acutely infected patients who successfully clear the infection, whereas a weak immune response leads to chronic infection and persistent liver cell injury, eventually leading to cirrhosis and hepatocellular carcinoma [22,23,24]. The stealth virus behavior that favors chronic infection as well as inflammation that causes acute and chronic hepatitis B are both influenced by the expression of proteins encoded by the precore/core region of the HBV genome [25].

The HBV nucleocapsid protein, core antigen (HBcAg), is a multifunctional protein that plays key roles in both viral lifecycle and the relationship between the virus and host’s immune system [26,27]. Within the nucleus of infected cells, HBcAg participates in the epigenetic regulation of the viral genome, interacting with supercoiled HBV-DNA, cccDNA, and host proteins and regulating the transcription of viral genes. Within the cytoplasm, HBcAg self-assembles into icosahedral viral nucleocapsids containing the HBV genome. HBcAg is strongly immunogenic, inducing antigen-specific T-cell responses which are critical for the immune control of HBV infection; however the role played by HBV nucleocapsid protein, particularly the secretory form, HBeAg, can change over time, favoring either the HBV immune evasion or eventually leading to the immune control of HBV infection. 

HBeAg is a non-structural protein translated from precore mRNA that is processed in the ER, secreted in the extracellular space, and circulated in the blood. HBeAg is not required for viral replication or infection; nevertheless, it plays a key role in the viral–host interplay and the establishment of chronic HBV infection [28,29]. The importance of this non-structural protein is demonstrated by the fact that it is conserved in all ortho-hepadnaviruses [30]. Since its discovery, HBeAg has been regarded primarily as an HBV accessory protein and used in clinical practice as ancillary marker of active viral replication [31]. Subsequent studies in both animal models and humans provided consistent evidence that HBeAg contributes to the establishment of viral persistence in the absence of inflammatory liver disease in vertical, mother-to-child transmission because of the pivotal role played by serum HBeAg, which crosses the human placenta [28,32,33]. The multifaceted immunomodulatory functions of HBeAg include the downregulation of TLR2, NF-κB activation, and IL-18-mediated signal of IFNγ expression, promoting viral replication. The analysis of T-cell responses to HBe/HBcAg in cord blood of HBeAg-positive and HBeAg-negative newborns reported no HBe/HBcAg-specific responses in T-cells derived from HBeAg-positive cord blood [34,35]. Accordingly, HBeAg expressed as a transgene in utero or 3 days after birth was shown to elicit T-cell tolerance to HBeAg and HBcAg [35,36]. Furthermore, maternal-derived HBeAg was shown to alter macrophage function in non-transgenic offspring, where viral persistence requires both maternal-derived HBeAg and the presence of HBeAg in the periphery [36].

Epidemiologic evidence pointed out that in the absence of prophylaxis, perinatal transmission of HBV is frequent when the mothers are HBeAg-positive (70–90% within 3 months), but significantly less frequent in HBeAg-negative mothers with lower viral loads (<10%) [37]. These evidence agree with previous clinical observations that acute fulminant hepatitis B occurs in newborns to HBeAg-negative, but not in those born to HBeAg-positive, mothers. Furthermore, HBV viral strains with consistent prevalence of mutations in precore and core promoter regions of the viral genome (HBeAg defective variants) were isolated from fulminant hepatitis B cases at any age [38,39,40,41]. Accordingly, primary infections with HBeAg-defective variants in both adults and neonates rarely become chronic, and their presence is associated with an increased risk of severe acute hepatitis [42,43,44]. The evidence that acute and even fulminant hepatitis B and viral clearance can occur in neonates infected with HBeAg-defective variants argues against the hypothesis that an immature immune system and liver microenvironment of newborns are responsible for chronic HBV infection without liver inflammation in children born from HBeAg-positive mothers. In actuality, HBV exposure in utero was demonstrated to induce complex changes in the newborn’s immune system, including an advanced immune maturation state or “trained immunity” with pronounced Th1 profile, but associated with the absence of HBV-specific T-cell responses [45]. By contrast, in young patients, it was shown that the immune profile of T-cell was not of tolerance, but characterized by an HBV-specific immune profile less compromised than that observed in older patients [46]. Thus, the old concept of immune tolerance versus immune activation/elimination HBV infection phases was more properly changed in that of not-inflammatory vs. inflammatory chronic HBV infection [47]. Nevertheless, age can only partially explain some of the epidemiologic and clinic–pathological reports [48], and there is compelling evidence in vitro and in vivo in both animal models and humans that the inflammatory switch of HBV infection parallels the dynamics of HBeAg expression prompted by significant variations of the wild-type/HBV-defective HBV ratio within the infecting viral quasispecies. The impact of HBeAg expression on virus persistence was confirmed in the animal model where the infection of neonatal woodchucks with wild-type woodchuck hepatitis virus (WHV) expressing WHeAg elicited chronic infection, whereas infection with a WHeAg-negative virus caused acute self-limited hepatitis in the newborns [49]. 

The relevance of HBeAg secretion in the natural history of HBV infection is also supported by the evidence that its production is modulated at both transcriptional and translational levels. Interestingly, all HBV genotypes that may differentially support the G1896A switch, creating a translational stop codon on the leader protein of HBeAg, may produce variants able to modulate HBeAg production, such as basic core promoter mutants or other mutants in the ATG of precore sequence, which are more frequently observed in genotype A, B, and C [50,51]. The G1896A mutation prevails in HBV genotype D because it confers a higher base-pairing stability of the stem loop of the encapsidation signal of the pregenomic RNA of this genotype. Such an advantage does not occur in genotypes A and H and to a lesser extent in genotypes B, C, and E [52,53,54], that, however, carry other mutations responsible for the lack of HBeAg production. 

In spite of a higher replicative fitness shown in vitro by HBeAg-defective HBV variants, serum HBV-DNA levels are lower in HBeAg-negative than HBeAg-positive CHB [55]. This impairment in virion productivity is not thought to be related to precore and basic core promoter (BCP) mutants, but to result from the virus/host’s immune system interplay [56,57,58,59,60]. The different replication fitness conferred by the different BCP/precore mutations according to infecting viral genotypes may explain the different natural history and epidemiology of HBV infection in geographical areas with different HBV genotype prevalence [61,62]. Accordingly, the longer-lasting HBeAg-positive infection in young females, before HBV vaccine implementation, was more frequent in areas with non-D HBV genotype endemicity, causing higher mother-to-child HBV transmission rate, persistent HBeAg-positive infection in newborns, and higher HBV endemicity in the same geographical areas. 

During chronic HBV infection, the reduced production of HBeAg is associated with major changes of intrahepatic HBcAg expression from only nuclear to both nuclear and cytoplasmatic that parallel the activation of anti-HBV immune response [14,19]. Acute fulminant HBeAg-negative hepatitis B was associated with an overwhelming B cell response, specific for HBcAg [63,64], and in HBeAg-positive immune activation or HBV clearance phases, HBe/HBcAgs were primary targets of the T-cell response [65,66,67]. Plasmacytoid dendritic cells pulsed with HBe/HBcAg-peptides stimulate T-cells derived from HBeAg-negative, but not HBeAg-positive, chronic patients [68]. Accordingly, in about one third of cases, the disease pattern of HBeAg-negative CHB is characterized by hepatitis exacerbations intervened by phases of complete normalization of serum transaminases which witness repeated, but ineffective, attempts at immune control of HBV replication [21]. The long-lasting ineffective immune elimination pattern induces the selection of HBeAg defective strains that become prevalent in the later phase of CHB natural history [69]. Thus, secretory HBeAg is tolerogenic, and cytosolic HBcAg is immunogenic and targets for HBe/HBcAg-specific CTLs once HBeAg-specific tolerance subsides (Figure 2). 

## 3. Natural Course and Antiviral Therapy 

Chronic HBV infection can be classified into five phases: (I) HBeAg-positive chronic infection, (II) HBeAg-positive chronic hepatitis B, (III) HBeAg-negative chronic infection, (IV) HBeAg-negative chronic hepatitis B, and (V) HBsAg-negative phase or occult HBV infection [70]. The natural history of chronic HBV infection is characterized by the sequential succession of these phases, and both their duration and phase to phase transition are modulated by the interplay between the virus and the antiviral host’s immune response. HBeAg-positive infection without virus-induced liver injury follows the induction of virus persistence and lasts till antiviral immune activation is triggered and the virus antiviral immune activation starts, leading to HBeAg-positive hepatitis B that can be short-lived with HBeAg to anti-HBe seroconversion and clearance of HBV replication, or last as HBeAg-positive CHB. The outcome of HBeAg to anti-HBe seroconversion is dual, depending upon the ability of the immune system to control HBV replication, and it can exit into HBeAg-negative infection without HBV-related liver disease or in HBeAg-negative CHB. The association of HBeAg-negative mutants with acute and mostly severe hepatitis B cases or hepatitis exacerbations in chronic HBsAg carriers and during HBeAg/anti-HBe seroconversion phases indicated a selective advantage for HBeAg-negative variants in such conditions. The introduction of precise and reproducible quantitative assays for the analysis of wild-type and HBeAg-minus HBV ratios in clinical specimens allowed study of the relations between the dynamics of HBV precore heterogeneity and the course of hepatitis B infection [69]. Accordingly, the relative dynamics of wild-type and HBeAg-defective HBV populations in children before and after HBeAg to anti-HBe seroconversion favor the hypothesis that the latter are selected after triggering of inflammatory liver disease [71,72], possibly because of their ability to escape the immune response once its activation has occurred and HBeAg immune tolerance is lost [73,74,75].

The selection of HBeAg-negative HBV during childhood is associated with the outcome of HBV infection in adults (Figure 3). 

A prospective study of precore HBV mutant dynamics and virological/clinical outcomes in 80 consecutive children (70 HBeAg-positive (87.5%), and 10 (12.5%) HBeAg-negative/anti-HBe-positive), mostly genotype-D-infected (91.2%), showed that the G1896A mutation appeared earlier and was selected during the HBeAg/anti-HBe seroconversion phase [72]. After HBeAg to anti-HBe seroconversion, eight children (14.6%) developed HBeAg-negative CHB, whereas 41 (74.5%) acquired an HBeAg-negative chronic infection and 6 (10.9%) lost serum HBsAg. In all children who were unable to clear HBV replication, HBeAg-minus HBV becomes prevalent in the circulating quasispecies, leading to HBeAg-negative CHB [72,73]. About 40% of them had additional mutations in the BCP region (G1764A), affecting HBeAg production at the transcriptional level [72]. On the contrary, the G1896A and BCP mutations were detectable only in 33 and 15% of carriers with HBeAg-negative infection, and 16% and none of those who cleared HBsAg during the follow-up, respectively. These findings, and the evidence in the adults that after HBeAg to anti-HBe seroconversion, a dominant G1896A precore mutant population was more frequent in patients with CHB (44.4%) than in HBeAg-negative infection carriers (19.6%), suggest that the faster and stronger the immune control of HBV replication is, the lower the selection of HBeAg defective variants [72,73,74,75]. Furthermore, the evidence that in HBeAg-negative infection carriers with dominant precore mutant population serum HBV-DNA levels were significantly higher than in those with dominant wild-type virus both in Caucasian and Asian patients [74,75] supports the hypothesis that the G1896A precore HBV variant behaves as a CTL escape mutant (Figure 1). 

HBeAg-negative CHB is a progressive liver disease that evolves into cirrhosis with 3–10% yearly rates, and spontaneous remissions are infrequent (<2%) [21]. The outcome is worsened by persistent viral replication, higher serum HBV-DNA levels, and the unremitting inflammatory disease profile (Figure 4) and once cirrhosis has developed by hepatitis exacerbations [21]. Standard interferon-alfa (IFN-α) was the first treatment option, and the same treatment schedules used for HBeAg-positive CHB (5–10 MU every other day for 16–24 weeks) were associated with high relapse rates (70–90%) in spite of about 70% treatment response [21,76]. Nevertheless, IFN-α reduced, by 2.5-folds, disease progression, and disease remission was more frequently observed in treated patients (14.6%) as compared to untreated (1.6%) [21]. Longer treatment courses (12–24 months) showed higher sustained response rate (22–30%) and 32–67% HBsAg loss in responders within 4–7 years post-treatment [76]. Serum HBsAg kinetics represents a useful guide for pegylated-IFN-α (peg-IFN-α) therapy, predicting the on-treatment response, as HBsAg decline from baseline correlates with 3 years post-treatment HBsAg loss and 10 IU/mL HBsAg levels at the end of treatment. This is associated with 52% HBsAg clearance probability at 3 years post-treatment follow-up, compared to only 2% in patients with higher levels [77]. HBV genotypes significantly influenced both baseline HBsAg levels and during-treatment HBsAg kinetics: the greatest differences between responders and non-responders were seen between weeks 12 and 24 in genotype A and baseline and week 12 in genotypes B and D infected patients, respectively [78]. During treatment, HBsAg and HBV-DNA declines proved useful to guide therapy at the single patient level with an early identification of non-responders [77,78,79]. Accordingly, the lack of any serum HBsAg decline combined with less than 2 log IU/mL serum HBV-DNA reduction after 12 weeks of peg-IFN-α identifies HBV genotype D non-responders with high accuracy and 100% negative predictive value [79]. 

Most of the HBeAg-negative CHB patients are currently treated with nucleos(t)ide analogs (NAs) that effectively inhibit HBV replication in most of the patients, who achieved undetectable serum HBV-DNA in >90% after 3 years of treatment, but serum HBsAg was cleared only in a minority of them (1%/year) [76]. Given the current lack of strong predictors of sustained virological response and the possibility of severe and life-threatening hepatitis B reactivations after NAs discontinuation, current EASL Clinical Practice Guidelines suggest stopping NAs treatment only in patients without cirrhosis, with at least 3 years of sustained virological suppression who can be followed with frequent and timely ALT and serum HBV-DNA testing for at least the first year after NAs withdrawal [70].

Thus, a functional cure of HBV infection (the clearance of both serum HBsAg and HDV-DNA) is achieved with the currently available antiviral treatments (peg-interferon-α and NA) in a limited number of patients with HBeAg-negative CHB, even if the continuous suppression of viral replication by NA halts liver disease progression, lowering the HCC risk and improving survival. A series of new antiviral drugs are currently under investigation in CHB patients, and in the near future, a personalized approach will exploit the potential of different drug combinations.

## 4. Diagnosis

In clinical practice, it is mandatory to distinguish HBeAg-negative infection from HBeAg-negative CHB, because the former is characterized by persistently low replicative levels of HBV in the absence of HBV-induced liver disease and shows an overall survival comparable to HBV non-infected individuals [70]. HBeAg-negative CHB is associated with median serum HBV-DNA levels above 20,000 IU/mL, but the fluctuating pattern of viremia makes the diagnostic HBV-DNA testing at a single time point inadequate for a stringent early differential diagnosis of HBeAg-negative CHB from HBeAg-negative infection, particularly because at least one third of patients with HBeAg-negative CHB show, as previously discussed, a viraemia pattern characterized by major fluctuations, with temporary declines of HBV-DNA below the threshold of 2000–20,000 IU/mL. During such phases of low replication, transaminases also normalize, leading to a profile similar to that of HBeAg-negative infection (Figure 4) [21]. Therefore, a prolonged period (at least 1 year) of every 3 months of testing of HBV-DNA is required and recommended by International Guidelines for an accurate differential diagnosis between the two conditions [70]. 

Alternatively, the single point combined testing of both quantitative HBV-DNA and HBsAg consistently improves the diagnostic performance in the identification of HBeAg-negative infection carrier in the case of HBsAg ≤ 1000 IU/mL and HBV-DNA ≤ 2000 IU/mL [80]. However, this holds true mainly for genotype-D-infected individuals, because of the major influence of HBV genotypes on HBsAg serum levels [78]. Recently it was shown that hepatitis B core-related antigen (HBcrAg) levels may accurately differentiate HBeAg-negative infection from CHB: in a cohort of 1582 European HBeAg-negative carriers, the threshold of 3.14 log U/ml showed an AUROC of 0.968 (% CI 0.958–0.977) with 91 sensitivity, 93% specificity, and 92.4% diagnostic accuracy in the identification of HBeAg-negative CHB, independently of HBV genotype [81]. The question remains about the long-term outcome of HBeAg-negative carriers with viremia > 2000 IU/mL, but persistently ≤ 20,000 IU/mL with normal serum transaminases. An accurate prospective study of 153 HBeAg-negative HBsAg-carriers with HBV-DNA ≤ 20,000 IU/mL with an overall follow-up of 5 years showed that viraemia persistently ≤ 20,000 IU/mL predicts a benign clinical outcome: associated with transition to HBeAg-negative infection in 43% of cases, whereas only 13.1% of them showed progression to HBeAg-negative CHB, usually during the first year of follow-up [82]. Thus, highly stringent criteria, including the combined quantification of serum HBV-DNA, HBsAg, and possibly HBcrAg, should be used to screen HBV carriers with HBeAg-negative infection to diagnose CHB at its early stages when current antiviral therapy can provide a complete cure halting disease progression before advanced liver fibrosis develops [76].

## 5. Conclusions

HBeAg-negative CHB emergence, persistence, and outcome stem from the dynamics of HBeAg production and expression, which are modulated by naturally occurring precore/BCP mutants whose replicative fitness vary between HBV genotypes. HBeAg-negative variants provide selective advantages over wild-type HBeAg-positive HBV behaving as escape mutants during the spontaneous attempts of antiviral immune clearance. Upon antiviral immune activation, the majority (about 60%) of HBsAg-positive individuals with HBV-induced inflammatory liver disease spontaneously achieve an effective control of viral replication with clearance of intrahepatic inflammation and transition to an indolent HBeAg-negative infection phase; some of them eventually clear circulating HBsAg. Nevertheless, even after HBsAg loss, HBV persists within some hepatocytes in the form of viral mini-chromosome, supercoiled covalently closed HBV-DNA (cccDNA), which eventually triggers the reactivation of viral replication in case of impaired immune competence. The more rapid and effective the immune control of HBV replication after the triggering of immune activation phase, the lower the chance for selection of precore HBV mutants which are associated with both origin and persistence of HBeAg-negative CHB in patients with inadequate immune control of HBV infection. Further studies are needed to understand the fine mechanisms of anti-HBV immune activation that triggers HBV-specific liver inflammation that leads to the selection of HBeAg defective mutants during chronic HBV-induced inflammatory liver disease. 

## Figures and Tables

**Figure 1 viruses-14-01691-f001:**
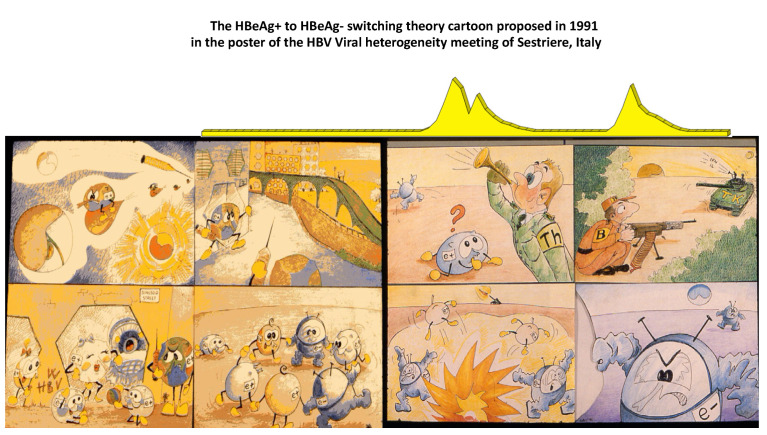
At the meeting on the Pathogenetic Implications of HBV Heterogeneity of 5–7 April 1991 in Sestriere, Italy, our group proposed, in a cartoon, depicted as poster, a hypothetical theory to explain the clinic–pathologic implications of the differential expression of HBeAg provided by wild-type HBeAg-positive virus and HBeAg-defective variant. The 4 pictures on the left side of the cartoon outline the induction of persistent HBV infection without liver injury favored by wild-type HBV population. The 4 pictures on the right side represent, in sequence from top to bottom, the induction of anti-HBV inflammatory response that leads to the selection of HBeAg-negative variants behaving as immune escape mutants.

**Figure 2 viruses-14-01691-f002:**
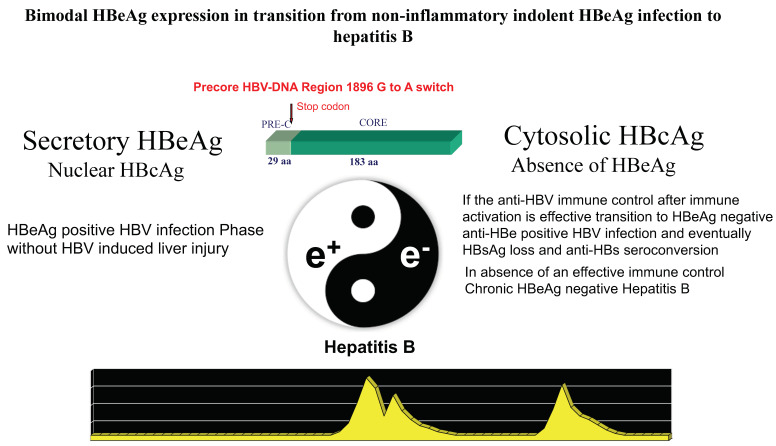
The dynamic change of the protein expression coded by the precore/core HBV-DNA region from the secretory HBeAg to the cytosolic HBcAg pattern conditions the interplay between HBV replication and the host’s immune response. The G1896A mutation inducing a stop codon that blocks the production of the precore leader sequence necessary for HBeAg secretion provides a major functional switch modulating the dynamics of HBeAg expression associated with HBV-induced liver inflammation during chronic HBV infection.

**Figure 3 viruses-14-01691-f003:**
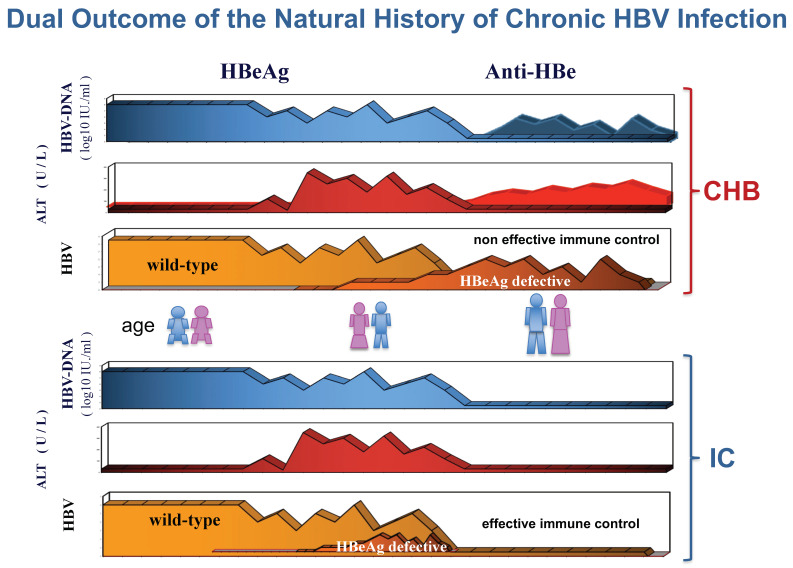
The HBeAg defective precore HBV variants are selected during HBeAg/anti-HBe seroconversion phase in children with poor immune control of HBV replication, leading to HBeAg-negative chronic hepatitis B (CHB) during adulthood in the case of defective immune control, while in the case of an effective HBV immune control, the outcome is HBeAg-negative infection without HBV-related liver disease (inactive carriers: IC).

**Figure 4 viruses-14-01691-f004:**
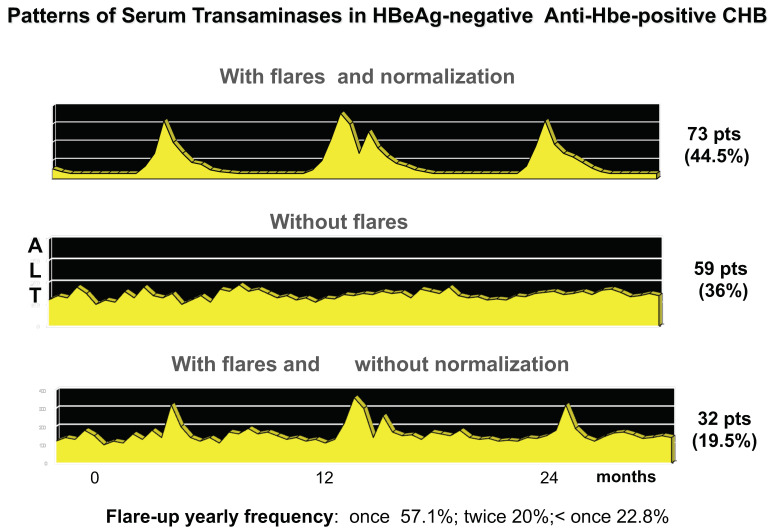
The natural course of HBeAg-negative CHB is characterized by different ALT patterns: hepatitis reactivations (mostly asymptomatic) intervened by long-lasting periods where the transaminase levels are normal, persistent mild to moderate ALT elevation, or persistent mild ALT elevation with superimposed occasional hepatitis exacerbation with highly elevated ALT (>500 IU/dL) [56].

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
