# Peer review of "HBeAg-Negative/Anti-HBe-Positive Chronic Hepatitis B: A 40-Year-Old History"

_viruses, 2022, doi:10.3390/v14081691_

Round 1

Reviewer 1 Report

This is a well written review that incorporates the contribution of Italian scientists to the discovery, study and treatment of HBeAg-negative/anti-HBe-positive chronic hepatitis B.

Some minor points to be addressed:

Lines 15-17  Please add some punctuation marks in this long sentence “Hepatitis B “e” Antigen (HBeAg) negative chronic hepatitis B (CHB), after 40 years since its discovery in the Mediterranean area, has become the most prevalent form of HBV induced liver disease worldwide and major health care burden caused by HBV infection.”

Lines 41,48, 51, 64, 74 What does “we” stand for? The 3 authors? It can be replaced by the name of the Italian group or the Italian scientists that you imply.

Line 41  Please add the reference for “Using this technique we found that  not only HBeAg-positive patients had serum HBV-DNA, but also HBeAg-negative/anti- HBe-positive patients with CHB without HDV infection.”

Line 201 Spelling mistake “citosolic” please correct to “cytosolic”

Author Response

In the new editing we included all the corrections suggested by the reviewer including the citation of all the specific reference manuscripts when we refer to our group works. Minor spelling mistakes were corrected. 

Reviewer 2 Report

Manuscript ID: viruses-1836903

Type of manuscript: Review

Title: HBeAg-negative/anti-HBe-positive chronic hepatitis B: a 40 years old 

history

Authors: Ferruccio Bonino* *

This is an informative review about HBeAg-negative chronic hepatitis B. 

Some sentences were too long to understand. Authors had better revise them. 

Minor; 

It is necessary to explain all abbreviations. For examples, ‘IC’ in Figure 3., ‘NPV’ in page 7, ‘CPG’ in page 7, etc. 

Author Response

We diete the new version with the help of an English language native translator. Abbreviations are now fully explained in the text. 

Reviewer 3 Report

Bonino F, et al. reviewed a 40-year-history of HBV infection, with special reference to HBeAg-negative chronic hepatitis. This review contains interesting and valuable information but has several issues to be addressed for publication.

 Major

I have a question regarding the authors’ description, “We hypothesized that wild-type HBV secreting HBeAg favors the induction of chronic infection whereas the HBeAg-minus variant, surges after the establishment of chronic infection and selectively prevail displacing wild-type virus once HBeAg is recognized as an immune target since in chronic inflammatory infection the lack of HBeAg expression provides a selective advantage to the virus (Figure 1).” (Line 64-68, p2)

It is now generally recognized that HBeAg acts as an immune decoy in the immune-tolerant phase. This concept is easy to understand why children delivered from HBeAg-positive mothers easily become HBV carriers. On the contrary, HBeAg-positive patients show a severe form of hepatic injury in the immune-active phase, compared with HBeAg-negative patients. This cannot be explained by the immune decoy hypothesis. Although this contradictive phenomenon was not solved yet, authors can discuss it.

 Minor

1.      Spaces or periods are wrongly used in several sentences. Please revise them,

2.      Please revise citosolic→cytosolic in Figure 2.

3.      Abbreviation should be mentioned in the Figure. Figure 3: IC (inactive carrier)

4.      HBeAg negative CHB should be HBeAg-negative CHB (Line 308, p8)

5.      Figure 4: Patterns of Serum Transaminases in HBeAg negative Anti-HBe positive CHB

6.      Please grammatically revise the following sentence. I cannot understand what the authors mean.

The more rapid and effective the immune control of HBV replication after triggering of immune activation phase the lower the chance for selection of precore HBV mutants which are associated with both origin and persistence of HBeAg-negative CHB 366 in patients with inadequate immune control of HBV infection. 

Author Response

The the tantalizing question raised by the reviewer as major pint remains open tin light of all the current knowledge and we completely agree. Accordingly this is extensively underlined in the discussion and conclusion. The new editing ameliorated the understanding of the different view points discussed.

Minor corrections have been done as suggested as well as a more easy to understand rephrasing.